

# Reviews and syntheses: Carbon vs. cation based MRV of Enhanced Rock Weathering and the issue of soil organic carbon.

Jelle Bijma[1], Mathilde Hagens[2], Jens Hammes[3], Noah Planavsky[4,10], Philip A.E. Pogge von Strandmann[5,6], Tom Reershemius[7], Chris Reinhard[8], Phil Renforth[9], Tim Jesper Suhrhoff[10,4], Sara Vicca[11], Arthur Vienne[11], Dieter Wolf-Gladrow[1].

[1]Marine Biogeosciences, Alfred-Wegener-Institut, Helmholtz-Zentrum für Polar- und Meeresforschung, D-27570 Bremerhaven, Germany
[2]Soil Chemistry Group, Wageningen University & Research, Wageningen, Netherlands
[3]Carbon Drawdown Initiative Carbdown GmbH, Fürth, Germany
[4]Department of Earth and Planetary Sciences, Yale University, New Haven, CT 06520-8109, USA
[5]Mainz Geochemistry and Isotope Centre (MIGHTY), Johannes Gutenberg University, D-55122 Mainz, Germany
[6]London Geochemistry and Isotope Centre (LOGIC), University College London, London WC1E 6BT, UK
[7]School of Natural and Environmental Sciences, Newcastle University, Newcastle upon Tyne NE1 4LB, UK
[8]School of Earth and Atmospheric Sciences, Georgia Institute of Technology, Atlanta, GA 30332-0340, USA
[9]School of Engineering and Physical Sciences, Heriot-Watt University, Edinburgh, EH14 4AS, UK
[10]Yale Center for Natural Carbon Capture, Yale University, New Haven, CT 06511, USA
[11]Biobased Sustainability Engineering (SUSTAIN), Department of Bioscience Engineering, University of Antwerp, 2020 Antwerp, Belgium

*Correspondence to*: Jelle Bijma (jelle.bijma@awi.de)

## Abstract

We discuss the "monitoring, reporting & verification" (MRV) strategy of Enhanced Weathering (EW) based on carbon accounting and argue that in open systems such as arable land, this approach is ill-suited to close the balance of all carbon fluxes. We argue for total alkalinity (TA) as the central parameter for the carbon based MRV of EW. However, we also stress that tracking alkalinity fluxes using a systems-level approach is best done by focusing on charge balance maintenance through time. We start by explaining the concept and history of alkalinity conceptualization for the oceans. The same analytical method first proposed for the oceans - titration with a strong acid - is now commonly used for porewaters in agricultural soils. We explain why this is an accurate analysis for ocean water and why it is unsuitable to record TA for porewaters in agricultural soils. We then introduce an alternative MRV based on cation accounting. This requires translation of "carbon currency" into "cation currency" based on the concept of the "explicit conservative expression of total alkalinity" (Wolf-Gladrow et al., 2007). We finally discuss the fate of cations released from the weathering of basalt, soil cation dynamics and close by suggesting open research questions.



## 1 Introduction

Relying on future atmospheric carbon dioxide removal (CDR) to avoid dangerous climate change while continuing to burn
35 fossil fuels has been termed the 'overshoot myth' and is extremely dangerous (e.g. Schleussner et al., 2024). The average
atmospheric $CO_2$ concentration reached 424.6 parts per million in 2024, 3.5 parts per million above 2023, and 52% above pre-
industrial levels (Friedlingstein et al., 2024). Kotz et al. (2024) estimate that, due to global warming, the world economy is
committed to an income reduction of 19% in the next 26 years. This equals $38 trillion which is six times higher than the
abatement costs of limiting global warming to no more than 2°C. Immediate emissions reduction through realisation of current
40 national pledges, and continued strengthening of these pledges, is the only realistic way to limit global warming to well below
2 °C above pre-industrial levels — the goal set by the 2015 Paris Agreement on climate change (Liu and Raftery, 2021; Ou et
al., 2021; Linow, 2022; Meinshausen et al., 2022; Dafnomilis et al., 2024; Ripple et al., 2024). Nonetheless, a consensus view
is that to meet this goal, billions of tons of CDR will have to accompany emissions reduction, given the prevalence of hard-to-
abate residual emissions (Luderer et al., 2018; Buck et al., 2023; Ipcc, 2023). For the 2 degrees target without overshoot, the
remaining carbon budget is around 1200 Gt $CO_2$ (Lamboll et al., 2023). It has been suggested that if we wish to return to a
safe climate for universal human habitation after reaching "net zero", including the global South, we need to further lower the
atmospheric $CO_2$ levels to below 350 ppm (Hansen et al., 2023). Hence, CDR will have to become a major industry for multiple
generations to come, in the order of the current fossil fuel industry. It is estimated that we will need to remove roughly 7–9
billion tonnes of $CO_2$ per year by 2050 to meet the Paris climate goals (Peplow, 2024). McKinsey research[1], approximates that
the value of the voluntary carbon market could grow to more than $50 billion by 2030 and that the CDR industry could be
worth up to $1.2 trillion annually by 2050. The annual burden towards 350 ppm depends on the adopted timescale.

Given the scale of the challenge, it is imperative to consider how long-term natural regulation of atmospheric $CO_2$
concentration (Appendix 1) may be harnessed to bring anthropogenic CDR to scale, both via the oceanic and the terrestrial
ecosystem. Although increasing and maintaining the amount of carbon stored in the biosphere and in soils is essential to
meeting climate goals, reservoirs such as these with a $CO_2$ storage period of less than 1000 years are insufficient for neutralising
long-term residual fossil $CO_2$ emissions under net zero emissions scenarios (Brunner et al., 2024).

In the absence of anthropogenic forcing, and on timescales of a few thousand years, variations in atmospheric $CO_2$
concentration[2] are driven by shifts in ocean circulation, the pelagic carbon cycle and the ratio of dissolved inorganic carbon
(DIC) to alkalinity[3] delivered to the oceans via river discharge. The resulting $CO_2$ is then partitioned between the atmosphere
and ocean. Without major carbon perturbations, this quasi steady state is modulated annually by the seasonality of the net
primary productivity on land.

---

[1] https://www.mckinsey.com/capabilities/sustainability/our-insights/carbon-removals-how-to-scale-a-new-gigaton-industry
[2] For "natural control of the atmospheric $CO_2$ concentration" see Appendix 1.
[3] For an explanation of 'alkalinity' see Appendix 2.





The current carbon perturbation by anthropogenic emissions exceeds the natural regulation speed of the oceanic buffering system (Maier-Reimer and Hasselmann, 1987; Hooss et al., 2001), and lowers its capacity (Hagens and Middelburg, 2016). Currently, ~25% of our annual $CO_2$ emissions are taken up by the ocean primarily by dissolution affecting the oceanic
carbonate chemistry and, to a much smaller extent, the ocean's "biological carbon pump" (e.g., Siegenthaler, 1993; Friedlingstein et al., 2023; Friedlingstein et al., 2024). More than 90% of the excess insolation energy ("global warming") is also taken up by the ocean (Venegas et al., 2023). While lessening the impact of climate change on terrestrial ecosystems, $CO_2$ and heat uptake comes at a high cost for marine ecosystems. Ocean acidification and heat waves (Frölicher et al., 2018) are now threatening marine life, notably coral reefs that are a hotspot for oceanic biodiversity. The effects of these perturbations
could be partially ameliorated through the addition of alkalinity to the ocean. One way of generating this alkalinity is via the addition of pulverised silicate rock on arable land ("Enhanced Weathering" or EW). Because it is estimated that 25% of the hard-to-abate-emissions will come from the agricultural sector (Edelenbosch et al., 2024), it would be proactive to already implement and scale up EW with the support of farmers. This can be achieved by developing business models based on carbon credits and the agronomic benefits related to EW that have been demonstrated in some studies (Manning and Theodoro, 2018;
Swoboda et al., 2022).

## 2 Enhanced Rock Weathering and agriculture

The addition of crushed silicate rock to (arable) soils, its subsequent dissolution, its reaction with atmospheric and soil $CO_2$, and the transport of the resulting weathering products to the ocean, constitute a promising CDR approach (termed 'enhanced (rock) weathering' E(R)W) (Schuiling, 1990; Schuiling and Krijgsman, 2006; Hartmann et al., 2013; Amann et al., 2020;
Beerling et al., 2020). Initial emissions associated with the supply chain of crushed rock are small (ca. 6-23% e.g. Renforth, 2012) in comparison to the overall carbon dioxide removal from the atmosphere that can be expected to result from weathering over time. Additionally, the infrastructure required for EW deployments at farms is the same as that already in use for standard agronomic practices, reducing barriers to implementation.

Subsidies and incentive structures for farming operations to implement specific practices are commonplace in most developed
countries, and increasingly in developing countries. Agriculture is already a system with extensive subsidy structures for encouraging CDR, via no-till agriculture to facilitate soil organic carbon storage – though currently, agricultural systems are a large net source of greenhouse gases (GHGs) to the atmosphere (Rosa and Gabrielli, 2023). A significant source of agricultural emissions[4] comes from the production/application of crushed carbonate rock (e.g., limestone) to raise soil pH via liming in settings where acidity is supplied by nitrogen fertilisers. Weathering of silicate rock can possibly replace liming as it will also
increase pH. In addition to deacidification, applying rock as a mineral fertiliser may provide co-benefits to farmers by supplying plant-available micronutrients to soils that are otherwise scarce in many depleted systems (Beerling et al., 2018;

---

[4] Note that without the addition of lime, acidity would still react with natural alkalinity pools. On multi-year timescales lime addition may become neutral or even negative when the calcium is charge balancing dissolved carbon (see section 6.1)



Haque et al., 2019; Vienne et al., 2022). For this reason, application of basalt and other mafic rocks to fields is an agronomic practice that has been carried out for centuries around the world, albeit at relatively small scale compared to modern farming methods (Swoboda et al., 2022). Overall, techno-economic and life-cycle assessments of EW deployments in various
geographies are generally favourable to the feasibility of EW to scale as a method of CDR (Renforth, 2012; Beerling et al., 2020; Kantzas et al., 2022; Zhang et al., 2023).

To allow CDR methods to scale rapidly but robustly, it is important to develop relatively simple monitoring, reporting and verification (MRV) approaches that are sufficiently rigorous to give a high degree of confidence in the veracity of carbon removal. A variety of MRV approaches are currently being developed, which either focus on quantifying weathering products
primarily in the aqueous phase but also secondary solid phases, or using soil-based approaches to estimate the fraction of rock powder that has dissolved (Almaraz et al., 2022; Clarkson et al., 2024). Most, if not all, aqueous-phase-based protocols are based on inorganic carbon accounting. This is due in part because of the requirement of permanent carbon storage, but mostly because of the assumption that the inorganic and organic carbon cycles in soils can be treated as two distinct systems.

Recently, several publications have stressed the importance of the organic carbon cycle on the net impact of CDR via EW, and
suggested a requirement to include the organic carbon pool into the MRV of EW (Klemme et al., 2022; Vicca et al., 2022; Yan et al., 2023; Linke et al., 2024a; Linke et al., 2024b; Manning et al., 2024; Niron et al., 2024; Vienne et al., 2024; Anthony et al., 2025; Lei et al., 2025; Vienne et al., 2025). Organic carbon fluxes are easily 10x higher than the inorganic carbon fluxes generated through EW. Because soils contain substantial amounts of organic carbon (SOC) it is critical to avoid SOC loss as much as possible or compensate them by the gain in soil inorganic carbon (SIC).

However, carbon accounting in an open system such as arable land is extremely difficult because of the efflux of $CO_2$ to the atmosphere and the unknown impact of "organic alkalinity" on the carbonate chemistry in the pore water. In a recent publication te Pas et al. (2025) conclude, on the basis of a short term soil column study, that 93 to 98% of the weathering products are retained within the soil profile and do not show up in leachate. This is in line with several other experiments that reported no or low leaching losses (Larkin et al., 2022; Vienne et al., 2024; Vienne et al., 2025) and modelling studies
suggesting lag times of up to several decades (Kanzaki et al., 2024). Hence, leachate-based total alkalinity (TA) measurements hugely underestimate weathering rates. te Pas et al. (2025) argue that soil-based mass balance approaches are more useful in quantifying weathering rates. Here, we build on the idea that cation accounting (Reershemius et al., 2023; Clarkson et al., 2024; Suhrhoff et al., 2024; Vienne et al., 2025) is a more practical and cost-effective MRV strategy, and equally as informative as accounting for all carbon species.

**3 Is alkalinity in soils different from alkalinity in the ocean?**

Alkalinity in the ocean can be expressed in terms of a cation surplus (i.e. cation "currency") or in terms of the anion gap that is filled by negatively charged carbonate species (i.e. carbon "currency"). Importantly, in the ocean both "currencies" have the same value but alkalinity in the ocean is traditionally measured and expressed in carbon "currency", using the gran titration



and Dickson standards. This is attractive as it is expressed in the same currency used for the carbon credit market and therefore
it has been the preferred currency also for pore water chemistry when monitoring EW in arable soils.

Formally, ocean TA is defined as the excess of proton acceptors over proton donors (see Appendix 2). The same definition can be applied to pore water in soils, but the outcome of the standard alkalinity titration is different because of differences in the chemical composition of soil pore water compared to seawater. TA in seawater is dominated by carbonate alkalinity ($[HCO_3^-] + 2[CO_3^{2-}]$).  Hence, for ocean water the outcome of Dickson's alkalinity titration almost equals carbonate alkalinity
which is almost equivalent to TA.

While for ocean water TA can be used in combination with either DIC or pH to calculate $[HCO_3^-]$, this is not the case for pore waters in the terrestrial realm, where $[HCO_3^-]$ variations in pore waters are dominated by the organic carbon cycle in the soil (as indicated by the high $pCO_2$ values). Song et al. (2023) found in coastal waters that "OrgAlk might modify $H^+$ concentrations by 3% – 69% (i.e., pH by 0.01–0.78)" and this may be even more pronounced in arable soils and thus also compromise pore
water pH. Hence, using TA and pH to calculate $[HCO_3^-]$ in leachate as a proxy for EW, will produce erroneous numbers.

Organic acids introduce other acid-base species that contribute to alkalinity. This implies that TA, as determined via titration, is not equal to "carbonate alkalinity" and can therefore not be used to calculate other carbonate system parameters. This can be understood when comparing the oceanic with the terrestrial carbon cycle. In the ocean, organic matter synthesised in the photic zone by primary producers flows into an efficient food web, mostly within the mixed layer, where most of the carbon
flows into higher trophic levels or is respired within the microbial loop. About 5-20% leaves the mixed layer and usually less than 1% ultimately ends up in deep ocean sediments. This is called the biological pump because it pumps carbon against the DIC gradient into the deep ocean where almost all organic matter is respired to $CO_2$. At steady state, the combined biological and solubility pumps resulted in an atmospheric $pCO_2$ of ca. 280 ppm in the Holocene.

By contrast, in the terrestrial system, the soil matrix keeps the organic biomass and the decomposition thereof to the site of
production and only dissolved inorganic (DIC) and organic carbon (DOC) will enter the groundwater. As a result of intense microbial respiration of particulate organic matter (POM) and the soil pore structure, soil $pCO_2$ is much higher than in the atmosphere (up to 4,000-40,000 ppm; Dietzen and Rosing, 2023) leading to a significant efflux into the atmosphere (Paessler et al., 2024; Paessler et al., 2025)[5], while the remaining POM resides in the topsoil. Many of those organic molecules are negatively charged and thus contribute to an "organic alkalinity" pool balancing cations, which as a first estimate can contribute
82.5-137.5 µeq $L^{-1}$ to TA in the standard titration procedure (see Appendix 3).

There is also growing consensus that dissolved organic matter (DOM) can significantly contribute to TA in coastal waters (Hemond, 1990; Cai et al., 1998; Ulfsbo et al., 2015; Kerr et al., 2021; Kerr et al., 2023; Wang and Cai, 2025). Ulfsbo et al. (2015) show that the measured excess alkalinity in the Baltic is consistent with an organic alkalinity derived from dissolved organic carbon, assuming that this dissolved organic carbon consists entirely of terrestrial humic substances. The contribution

---

[5] Note that these are blog posts that have not formally been reviewed but as they describe the results from the largest greenhouse based long term EW experiment worldwide and are prepared with the help of scientists, we report them here.





of polydisperse materials such as humic substances to titration alkalinity complicates Dickson's operational definition of titration alkalinity. In principle, organic acids could be accommodated by Dickson's TA definition if the pK values of these acids are known. The problem is that we usually do not know which organic acids contribute to TA and that their pK values are not discrete as for the carbonate or borate systems but are rather a polydisperse distribution of many pK values (Ulfsbo et al., 2015). It is therefore concluded that titration alkalinity should currently not be one of the parameters used to characterise

the $CO_2$ system in organic-rich waters, such as pore waters from agricultural fields. The organic alkalinity, e.g. in the Baltic, typically originates from riverine and groundwater fluxes derived from such fields. As mentioned before, this is not surprising because agricultural fields can be viewed as organic carbon reactors where, next to above ground biomass production, charged organic molecules such as e.g. humic acids are produced and microbial decomposition and humification produce charged POM and DOM. Consequently, the contribution of organic alkalinity can be significant and needs to be considered for carbon based

MRV of EW in agricultural settings. To account for this, methods have been developed to quantify organic alkalinity directly (Kerr et al., 2023; Song et al., 2023; Wang and Cai, 2025). However, these methods are currently not used in carbon based MRV protocols (such as those proposed by Isometric (Sutherland et al., 2025) and puro (https://puro.earth/puro-standard-carbon-removal-credits)).

For carbon accounting, usually two of three parameters (DIC, pH and alkalinity) are determined in the leachate to calculate

[$HCO_3^-$] flux, the target parameter with regard to permanent carbon storage in the ocean. However, the presence of a significantly negatively charged organic carbon pool, not only renders the Gran titration ineffective (as it does not result in the quantification of carbonate alkalinity) but also compromises pH (Song et al., 2023). Because of the high $pCO_2$ in pore waters, it is hardly possible to extract a sample for DIC analysis without equilibration to ambient conditions. As a consequence of the change in pH, [$HCO_3^-$] and [$CO_3^{2-}$] cannot be calculated with any significance.

As life is carbon-based, the DIC leaving the field, entering the groundwater and the river will be recycled many times in the food-chain before it finally enters the ocean, probably much later than the cations. Hence, the additional carbon finally stored in the ocean due to EW is almost certainly not the bicarbonate from the leachate in the field, but pre-existing DIC in the acidified ocean that is charge-balanced by the cations from rock weathering on the field. Importantly, the cation flux to the ocean will drive the same shift in radiative forcing as the equivalent removal of $CO_2$ via the production of bicarbonate on the

field.

## 4 Impact of ERW on organic carbon cycling

The addition of rock-flour to a soil leads to the release of cations that affect the chemistry and the biology of the soil. As a consequence, the dynamics of both the organic and the inorganic carbon fluxes between the different pools will change, which may lead to an increased carbon storage in long-term reservoirs, as also seen for liming in mineral soils (Wang et al., 2021;

Zhang et al., 2022).

Soil mineral amendments for enhanced weathering alter key soil properties such as pH and nutrient availability, stimulating microbial activity, and thus organic matter decomposition and SOC dynamics (Vicca et al., 2022; Buss et al., 2024). For





example, Yan et al. (2023) observed that wollastonite addition significantly increased soil pH and DOC, leading to a 330% increase in soil $CO_2$ efflux across 12 land-use types (see also Paessler et al., 2024; Paessler et al., 2025). Xu et al. (2024) reported raised pH and microbial activity, with corresponding increases in soil enzyme activities and $CO_2$ emissions, after basalt application. Surprisingly, the $CO_2$ fluxes to the atmosphere and the increase of alkalinity in the leachate was more dependent on soil type than on the feedstock itself, suggesting a strong control by biology and soil composition (Paessler et al., 2024; A notable exception is steel slag, probably because it is extremely reactive).

On the other hand, a major outstanding question is whether increased $CO_2$ efflux simultaneously increases the residence time of the remaining SOC as a result of soil mineral amendments. Increases in so-called mineral-associated organic matter (MAOM) have been reported (e.g. Bai and Cotrufo, 2022; Chen et al., 2024). In recent papers (e.g. Ramos et al., 2024; Steinwidder et al., 2025 PREPRINT (Version 1)) the relationship between silicate weathering and organic carbon burial is discussed and concluded that clay mineral formation is the principal modulator of weathering fluxes and OC accumulation in soil. Hence, silicate addition may also stimulate SOC stabilization. Buss et al. (2024) also emphasized the role of primary silicates, such as basalt, in promoting SOC stabilization by forming MAOM and macroaggregates. In line with this, Vienne et al. (2024) observed reduced decomposition and $CO_2$ release following basalt application, but this effect depended on the presence of earthworms. Through cation tracing, Niron et al. (2024) found that basalt amendments enhanced the formation of secondary minerals such as clays and oxides. They also observed a reduction of the soil $CO_2$ efflux by approximately 2 tons $CO_2$ ha$^{-1}$, suggesting stabilization of SOM.

Increases and decreases in organic matter decomposition with mineral amendments have been reported, probably caused by soil type and properties, plant activity, microbial communities and interactions, indicating the context-dependency of the processes involved. Moreover, short-term effects on organic matter decomposition may be completely different from the long-term impact.

Only few experiments have explored the impact of EW on SOC dynamics, and these experiments were typically short in duration (weeks or months). However, time is expected to play a crucial role in shaping the balance between decomposition and stabilization. Stabilization effects, particularly through the formation of MAOM, may become increasingly important over time as mineral weathering progresses (Vicca et al., 2022).

## 5 MRV in agricultural soils

Closing budgets may be very difficult in open systems such as arable land and extremely cumbersome and expensive. Budgets can be carbon based (i.e. the "true" currency for carbon credits), or based on cation accounting. As stated before, both can be viewed as a different currency for alkalinity but are interchangeable. In the ocean, where the carbon is finally and permanently



stored, the ratio is roughly[6] equal to one (one mol carbon for one mol cation charge), although carbonate buffering likely reduces this to 0.7–0.9 (Renforth and Henderson, 2017).

For porewaters on land the situation is completely different and is expected to be strongly context-dependent. Continuously
monitoring all of the very dynamic organic and inorganic carbon sources and sinks in such an open system, especially in the first few years, is very difficult and expensive as it requires continuous monitoring. Therefore, it may be almost impossible to close the budget. Uncompetitively high requirements and costs for carbon based MRV compared to other approaches may impede the upscaling of EW. Here we argue that the MRV on the basis of cation accounting is more practical and efficient, complemented by monitoring SOC.

We propose a cation-based MRV that is related to the solid phase as it is much easier to handle in an open system and does not require continuous monitoring of carbon leaving the soil as biomass, in leachate or as a flux into the atmosphere. The "explicit conservative expression for total alkalinity" formulated by Wolf-Gladrow et al. (2007) is ocean-based but can easily be adopted for farmland and related to the biogeochemical processes in agriculture (e.g. impact of fertiliser/manure and liming). Both TA and DIC are conservative quantities with respect to mixing, changes in temperature and pressure and are therefore
used in (oceanic) carbon cycle models. For both the ocean and terrestrial (pore)waters it is thus important to understand changes in TA due to various biogeochemical processes.

It is virtually impossible to derive the impact of processes such as liming (dissolution and precipitation of carbonate) or the formation and remineralization of organic matter (what agriculture is actually about) or the addition of fertiliser or manure from the common expression for TA in terms of concentrations of non-conservative chemical species ($HCO_3^-$, $CO_3^{2-}$, $B(OH)_4^-$,
$H^+$, $OH^-$, etc.). Hence, Wolf-Gladrow et al. (2007) derived an expression for explicitly conservative total alkalinity (TAec; see Appendix 2) in terms of the total concentrations of major ions ($Na^+$, $Cl^-$, $Ca^{2+}$ etc.), and the total concentrations of various acid-base species (total phosphate etc.) from Dickson's original expression of TA under the constraint of electroneutrality. Changes in TA by various biogeochemical processes are then easily derived from the "explicit conservative expression" for TA because each term in this expression is conservative and obeys a linear mixing relation (Soetaert et al., 2007; Middelburg
et al., 2020).

In practical terms, the method proposed by Reershemius et al. (2023) for cation budgeting can be used in the field, provided soil heterogeneity in cation concentration can be accounted for to detect both feedstock addition and dissolution at relevant signal to noise ratios (Suhrhoff et al., 2024). A big advantage is that continuous sampling is not required but intervals of one or two years and can be synchronised with the sampling that farmers carry out to analyse soil parameters such as nitrate and
soil pH. Below the depth of typical soil sampling, the budgeting for cations can only be done via models, just as for carbon.

---

[6] Note that a change in TA does not lead to exactly the same amount of DIC change as suggested by simple stoichiometry (cf. the calculation in Köhler, P., Hartmann, J., and Wolf-Gladrow, D. A.: Geoengineering potential of artificially enhanced silicate weathering of olivine, Proceedings of the National Academy of Sciences of the United States of America, 107, 20228-20233, 10.1073/pnas.1000545107, 2010.).





Once the "cations" reach the ocean, the "cation currency" can be exchanged for the true "carbon currency" of carbon credits at an "exchange rate" of roughly one mol carbon for each mol of charge using Wolf-Gladrow et al. (2007). A suitable efficiency factor could be attributed to account for losses associated with carbonate buffering potentially between 9 - 30% (Renforth and Henderson, 2017; Kanzaki et al., 2023; Zhang et al., 2025).

**6 Fate of the cations**

Cations released by rock weathering end up in many different fractions, mostly associated with a temporal/transient storage of carbon via interactions with the cation exchange capacity (CEC) (Dietzen and Rosing, 2023; Zhang et al., 2024). As a result, most cations are temporarily held back before they reach the "depth of no return". At some point, cations are expected to start leaking through (in geochemical terms the breakthrough is reached). The cation-flow will most likely always be retarded, but
the breakthrough time of the system may vary as more cations are added on top depending on e.g. soil properties, rock characteristics, secondary mineral formation etc. It seems likely that the soil's cation "capacitor" also becomes saturated. The composition of the cations leaving the "capacitor" are very likely dependent on pH and water flow. Kanzaki et al. (2024) argue that the temporal disconnect between deployment of EW and climate-relevant CDR may have significant implications for the integration of EW into (voluntary) carbon markets. The question therefore is, if only those cations should be accounted as a
"carbon credit" that will directly and/or finally and permanently neutralize bicarbonate in the ocean? In the following we discuss the different pathways that cations can take, how they can also stabilize carbon on climate relevant timescales in transient SOC pools, and what the implications of cation loss for carbon fluxes are. The impact of agricultural practices, such as liming and fertilisation, on cation dynamics are briefly discussed in Appendix 4.

**6.1 Element exchange**

Both soil organic matter as well as clay minerals contribute to ion exchange and will be discussed below in separate sections. Soils have both an anion exchange capacity (AEC), and a cation exchange capacity (CEC) (e.g. Bergaya et al., 2013). The exchange capacity adsorbs anions or cations onto mineral surfaces (particularly clays or oxides), but also allows the continuous exchange with other negatively or positively charged ions in the soil pore water. In most soils, the CEC is much greater than the AEC, but this can change in highly weathered soils (e.g. Weil and Brady, 2017). CEC comprises both a permanent part
(due to isomorphic substitution in 2:1 phyllosilicate minerals) and a pH-dependent part composed of functional groups on edges of clay minerals and oxides and functional groups on organic matter. Under low pH conditions[7], the pH-dependent CEC will decrease as protons neutralise the negative charge of functional groups such that the AEC may become dominant in the pH-dependent part.

From the perspective of enhanced weathering, the ions (of both charges) liberated during silicate weathering (including
atmospheric-derived bicarbonate), will at first be taken up by the soils' exchange capacity. This may, depending on "base

---

[7] Note that low pH conditions are undesired in agriculture and usually antagonized with liming



saturation" and "spare capacity" also cause preexisting ions to desorb. These will then re-adsorb further down in the soil, causing a cascade reaction that retards the breakthrough of both anions and cations to groundwater, or indeed the pore water measuring point. Observationally, this process can take several years, and will depend on the soil characteristics (e.g. amount of clay), pore water pH, and water flow rates. Breakthrough times will also vary according to element (e.g. Pogge Von

Strandmann et al., 2019). These downward exchange reactions will likely still occur even when the CEC is "filled" with amendment-derived ions, so that new ions will always be retarded in their arrival, which could create some issues in a solution cation-based MRV method. For equal molar concentrations ions sorption preferentially occurs in the order: $Al^{3+} > H^+ > Ca^{2+} > Mg^{2+} > K^+ > NH_4^+ > Na^+$ (and desorption vice versa), but this is also concentration and pH dependent and varies for different clay minerals (e.g. Farrah et al., 1980; Gislason et al., 1996; Doi et al., 2020).

It should be noted that cations can also adsorb to organic compounds such as humic substances – the CEC complex is not composed purely of inorganic clays. The definition of humic substances is largely operational and is rooted in the history of soil science (Stevenson, 1994). "Humic substances" is an umbrella term covering humic acid, fulvic acid, humin, hydrophilic acid, etc. In terms of chemistry, they represent a continuum of humic molecules, constructed from similar (poly)aromatic, aliphatic, and carbohydrate units and containing the same functional groups (mainly carboxylic, phenolic, and ester groups),

albeit in varying proportions (Ghabbour and Davies, 2001). The presence of carboxylate and phenolate groups gives these substances the ability to form chelate complexes with ions such as $Mg^{2+}$, $Ca^{2+}$, $Fe^{2+}$, and $Fe^{3+}$ (Tipping, 1994), which is an important aspect of the biological role of humic acids in regulating bioavailability of metal ions to crops. In fact, depending on pH, humic substances account for 50 – 90% of the CEC in soils. The chelates release the cations lower in the soil profile, leading to podzol formation. But chelation is also important to speed up weathering, as it avoids secondary mineral

precipitation at the mineral surface.

When relating cation to carbon fluxes, it should be assumed that cations that are lost from solution to exchangeable sites will cause a charge-equivalent loss of bicarbonate due to re-equilibrium of the carbonic acid system (e.g. Kanzaki et al., 2024; Sutherland et al., 2025). If base cations adsorption is charge balanced by a release of protons (i.e., increase in base saturation), the resulting released acidity may react with bicarbonate and cause degassing of $CO_2$. The CDR associated with the cations

lost to the CEC complex may be realized at a later point in time when base cations are released from CEC sites due to a decrease in base saturation (Kanzaki et al., 2024).

## 6.2 Clay mineral formation

Secondary minerals that form from silicate weathering (oxides, clays, zeolites) are ubiquitous during weathering. Initially, rainwater is undersaturated with regard to primary silicate minerals (olivine, pyroxene, feldspar, etc.). However, the dissolution

of these minerals drives secondary minerals to supersaturation, and their precipitation then maintains primary mineral undersaturation, which allows these minerals to carry on dissolving. Rainwater is initially undersaturated for all silicate minerals (primary and secondary), so the first step of dissolution is "for free", but then secondary minerals will also start




precipitating. Secondary mineral formation may have direct effects on enhanced weathering processes, such as a physical change of field drainage patterns or surface passivation, where the relatively unreactive secondary minerals coat primary

mineral grains, slowing reaction rates (e.g. Beerling et al., 2020; Calabrese et al., 2022). Secondary minerals also cause chemical changes. Principle among these is the effect secondary minerals have on $CO_2$ sequestration. Simply put, secondary minerals can retain cations that would otherwise be used to sequester $CO_2$ by balancing carbonate or bicarbonate ions in pore waters. Thus, at first glance, secondary mineral formation seems to make $CO_2$ sequestration via silicate weathering less efficient; or, to put it another way, clay formation is a source of $CO_2$ (e.g. Pogge Von Strandmann and Henderson, 2015;

Kalderon-Asael et al., 2021; Pogge Von Strandmann et al., 2021). However, algae in soils and streams preferentially leach nutrients from secondary minerals (Grimm et al., 2019), and this attachment to clay particles therefore helps to bury the associated organic carbon(Kennedy and Wagner, 2011; Kennedy et al., 2014). When fresh water enters the ocean, flocculation of clays and associated organic matter occurs. Thus, while clays hinder $CO_2$ sequestration via the inorganic pathway which is the primary goal of ERW, they may assist $CO_2$ sequestration via the formation and burial of organic carbon.

When observing element mobility in basaltic weathering, K and Na mostly stay in solution while, e.g. Fe and Mn mostly go into secondary minerals. For example, in a basanite from the Eifel region, Ca and Mg have mobilities around 5x less than Na, suggesting that 80% of Ca and Mg are going into clays. In basaltic rivers, about 70% of Ca is going into clays. Figure 1 summarizes estimates of the proportion of Ca and Mg from weathered basalt ending up in secondary precipitates (notably clay minerals) from several studies. It shows that the percentage of cations retained by secondary mineralisation is highly variable

but a dominant process that needs to be taken into account with regard to the MRV of EW.

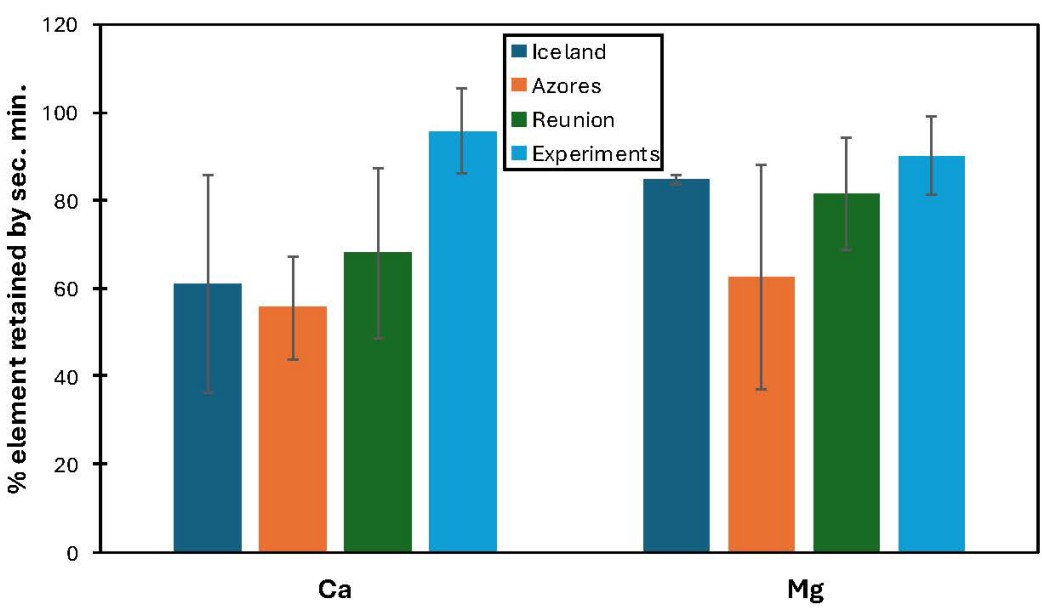



**Fig. 1: Estimates of the proportion of Ca and Mg from weathered basalt ending up in secondary precipitates (notably clay minerals), for Iceland (Pogge Von Strandmann et al., 2006), Azores (Pogge Von Strandmann et al., 2010), Reunion (Louvat and Allègre, 1997). The basalt weathering experiments are from Pogge von Strandmann et al. (2019). The overall method of the mobility calculation comes from Gislason et al. (1996). The error bars are the variability in the different rivers.**

Amorphous precursors of clay minerals, such as imogolite ($Al_2SiO_3(OH)_4$) and allophane ($Al_2O_3 \cdot (SiO)_{1.3-2} \cdot (H_2O)_{2.5-3}$), form very quickly and are characterised by a large specific surface area with variable and permanent surface charge, having a great affinity for organic functional groups. Basile-Doelsch et al. (2007) show that these amorphous precursors can store large amounts of organic matter which turns over very slowly and plays a pivotal role in the storage of a transient organic carbon pool on geological timescales beyond 100ky.

Hence, secondary minerals contribute significantly to the stabilisation and build-up of SOC (mineral associated organic matter or MAOM). Recent work (Niron et al., 2024; Steinwidder et al., 2025 PREPRINT (Version 1); Vienne et al., 2025) demonstrates that sequential extraction methods are powerful tools to understand the role of cations and secondary mineral formation in SOC stabilisation. Some conditions (e.g. high pH) will favor carbonate formation, while other conditions, such as low water flow rate lead to oversaturation and clay formation or cation bridging with SOC (Rowley et al., 2018; depending also on the type of OM). Song et al. (2022) conclude that while Fe minerals (e.g. pyroxenes) play an important role in stabilizing SOC, the electron transfer role of Fe also exerts a crucial influence upon SOC turnover. For instance, the coupling between $Fe^{3+}$ reduction with SOC oxidation can increase the efflux of $CO_2$.

In many enhanced weathering studies secondary mineral formation has been neglected, as its formation rate was considered to be too slow and measuring clay formation (as opposed to clay presence) has traditionally been difficult. "Non-traditional" isotopes, such as those of lithium or magnesium, can however measure and quantify secondary mineral formation (e.g. Pogge Von Strandmann et al., 2021). Using these methods, studies have shown that clay formation is often rapid, with experimental studies showing clay formation within a few hours (Jones et al., 2012; Pogge Von Strandmann et al., 2019; Pogge Von Strandmann et al., 2022a; Pogge Von Strandmann et al., 2022b; Pogge Von Strandmann et al., 2025) and natural groundwater studies showing clay formation within 1–2 months (Oelkers et al., 2019). This can be understood as primary silicate minerals cannot continue to dissolve without the reaction products being washed away or incorporated into secondary minerals.

However, the amount of secondary mineral formation per unit of silicate rock dissolved varies according to different parameters. Natural river studies have shown under which conditions secondary mineral formation is maximised. This largely depends on the water-rock contact time, which in turn is controlled by the discharge (e.g. Dellinger et al., 2015; Wilson et al., 2021; Zhang et al., 2022; Pogge Von Strandmann et al., 2023), and "shielding" by previously formed secondary minerals (Krause et al., 2023). Thus, when discharge is high, water-rock contact times are low, and secondary mineral formation is minimised. Conversely, when discharge is low, secondary mineral formation is relatively high. This means that in rapidly draining soils with a high porosity, the $CO_2$ drawdown efficiency by silicate weathering may be higher. Repeated amendment with silicates may change the porosity however, and thus diminish the $CO_2$ sequestration efficiency. In conclusion, cations





stored in secondary minerals cause an equivalent loss of $CO_2$. The timescale over which this happens and the fraction of loss is variable (and can therefore be optimized for).

### 6.3 Soil Organic Carbon (SOC) and Dissolved Organic Carbon (DOC)

Potential increases in SOC are a bonus on top of inorganic CDR – but a very interesting one, as the stabilisation of SOC via cation-bridging (e.g. Rowley et al., 2018) and association with minerals (MAOM; Chen et al., 2024), stores ca. 5 (Matus et al., 2014) to 10 times (Bai and Cotrufo, 2022) more carbon per unit charge/alkalinity than bicarbonate. Chen et al. (2024) call this the mineral carbon pump. However, since it is just a transient storage[8], those cations should only be counted as "one charge finally balancing one bicarbonate ion", just as cations reaching the ocean. When MAOM is decomposed, a small part of the
cations may or may not end up in bacterial necromass. The elemental composition of microbial biomass may vary between $C_4H_7O_{1,5}N$ to $C_5H_8O_{0,8}N$ with small amounts of P and S and trace amounts of Fe, K, Mg, Ca (Kästner et al., 2021). It is therefore safe to argue that the ratio of carbon per unit charge is much larger than unity and therefore much higher than for the final storage of carbon in oceanic carbonate alkalinity. Note that the remainder of the respired cations is not lost, but free to, either balance bicarbonate or to stabilise fresh SOC again. The bottom line is that the carbon/charge ratio during transient
storage will not go below unity as in the final permanent storage in the ocean and therefore that SOC should be accounted for on the basis of one mol carbon for one mol of cation charge.

### 6.4 The crop

Part of the cations will be taken up as micronutrients by the crop (e.g. Zn, Si, Fe, etc.) and potentially contribute to productivity and increased resistance to drought, frost, disease, and pests. Base cation charges in the crop are probably small in comparison
to the soil. Vienne et al. (2025) report that base cation charges in the plant pool were about two orders of magnitude smaller than in the soil pool (for maize plants).

If the crop is e.g. fodder for cattle, most of the cations will be returned to the (same or another) field as manure (minus a small percentage remaining in the cattle). If the crop (or the beef) is for human consumption, most of those cations will quickly end up in sewage. Assuming that the cations are not retained in the sewage treatment plant, most of the cations will be released to
a river and finally reach the ocean even before those taking the route via the groundwater from the fields. Hence in terms of carbon removal potential of EW, these cations are also still "in the game" and should be accounted for in the MRV provided

---

[8] Note that MAOM may be stable for centuries but is sensitive to disturbance.





the considered timescale is long enough for biomass to be decomposed and for the contained cations to once more charge balance bicarbonate.

Precise accounting may be difficult but in practice could be calculated as a percentage, reflecting the conversion efficiency of the food chain[9], i.e. the relative amounts of cations remaining in the organism (cattle, pork, chicken, etc.). For other products such as oilseed rape, similar calculations can be imagined, but finally most of the food produced will go through human stomachs or end up as compost which will finally be returned to the soil.

## 7 Conclusion and outlook

It is important to realise that the organic and the inorganic carbon cycles in agricultural fields are intimately related and cannot be separated for the MRV of EW. The idea that only the bicarbonate flux should be accounted for with regard to a carbon credit system is wrong as the impact of EW on a transient SOC pool can be significant and is stable.

It is correct to use total alkalinity (TA) as a master variable to monitor the efficacy of EW but the Gran titration that is currently applied to leachate waters to calculate the bicarbonate concentration is not suited as it does not include organic alkalinity. Furthermore, analysing and monitoring the carbon flows to close the budget is very hard to realise in an open system and requires continuous monitoring.

We propose to apply the explicit conservative expression for TA (TAec) in terms of the total concentrations of the major ions. The association of a major part of the cations with soil organic carbon via clay minerals may be transient but the stabilisation and storage is on climate-relevant timescales and just as effective as balancing DIC in the ocean. The carbon sequestration per cation charge is higher for organic- than for the inorganic-pathway. Even though most of the carbon associated with the additional cations from EW will finally be respired, the cations will stabilize fresh SOC and/or end up balancing carbonic acid or nitric acid from fertilisation. Hence, while the SOC pool is transient, the SOC concentration will likely increase and become higher than before EW.

Clay mineral formation can play a dominant role in the overall CDR by EW (Pogge Von Strandmann et al., 2025). A substantial fraction of the cations dissolving from the crushed rock may relatively quickly turn into clay mineral precursors, which can have a great affinity for organic functional groups. Hence, most of the weathering products are retained within the soil profile and do not show up in leachate. Although clay mineral formation reduces the CDR potential for carbonic acid, it likely does not lower the overall CDR. The association of clay minerals and their precursors with SOC can even store more carbon per unit charge, potentially creating a bonus.

We make the case that a cation-based MRV is much more practical and reliable than a carbon-based MRV. It does not require continuous monitoring and can be combined with the (bi)annual soil sampling that is carried out anyway. Because of its simplicity and economic benefits, it may be favourable for the upscaling of EW as a natural NET. For the cation-based MRV

---

[9] Vienne et al. (2025) report that base cation charges in the plant pool were about two orders of magnitude smaller than in the soil pool and can therefore be neglected in the balance.



we propose to sample the soil once a year, following the method proposed by Reershemius et al. (2023). The maximum CDR potential through cation liberation from dissolving rock can be calculated by stoichiometric analysis of the feedstock. The loss of cations from this is minimal and the majority of the cations are available for CDR.

Minor CDR losses from EW can be attributed to cation loss to crops. When used as fodder, the cations will mostly be returned via manure but the loss should be subtracted from the carbon credits for the farmer as he/she sells the produce. Finally, it should be noted that variability on the plot scale will be averaged by the law of large numbers, when EW is scaled up.

For the overall MRV several follow-up research questions arise:

- What is the average ratio of carbon per cation charge in clay minerals? How does that relate to the average annual
increase in SOC?
- What is the elemental composition of different crops? How much cations per hectare, in which stoichiometry, need to be subtracted from the CDR potential of the feedstock?
- What happens to cations in sewage treatment plants? How much and in what stoichiometry are they retained?

## 8 Acknowledgments

JB wants to express his deep gratitude to the Paessler Family Foundation for their continued support over the last 5 years. JB specifically wants to thank Dirk Paessler and Ralf Steffens for initiating the Carbon Drawdown Foundation, their hard and innovative work, both in the field and in the greenhouse, and for putting EW on the CDR map. Without their continued support, the research leading to the ideas developed in this paper would not have been possible. JB also wants to thank Ralph Koczwara (Hemmerbach/reverce), Klaus Gumpp (Alvacom/reverce) and Jan Lohse (reverce) for initiating "reverce" to tackle issues
specific to up-scaling EW. TJS acknowledges funding from the Swiss National Science Foundation (Grant P500PN_210790) as well as the Yale Center for Natural Carbon Capture.

## 9 Appendix 1: Natural atmospheric $CO_2$ control in a nutshell

On (geological) timescales much longer than a few thousand years, the atmospheric carbon dioxide concentration is controlled by the balance between $CO_2$ sources (volcanic degassing, metamorphism, rock organic carbon oxidation and potentially
carbonate weathering by sulfuric acid in mountain ranges) and sinks such as burial of organic matter and the burial of carbonate minerals. Both of these sinks are almost entirely driven by weathering, as it provides alkalinity (see Appendix 2) and cations such as $Ca^{2+}$ or $Mg^{2+}$ for carbonic acid neutralisation and carbonate formation (Walker, 1981), as well as nutrients that fertilise the formation of organic carbon and clay minerals that help to bury that organic matter (Kennedy and Wagner, 2011; Grimm et al., 2019). The oceans and the hydrological cycle play a pivotal role in partitioning this $CO_2$, as $CO_2$ not only dissolves in
water but also reacts with it to form bicarbonate ($HCO_3^-$) and carbonate ions ($CO_3^{2-}$). More than 99% of the dissolved inorganic carbon (DIC) in the ocean is in its ionic form. The reactivity of $CO_2$ with water is a major reason why the total dissolved carbon content of the ocean (DIC) is ca. 60x larger than the total $CO_2$ concentration in the atmosphere. Consequently, the carbonate chemistry of the ocean controls the $pCO_2$ of the atmosphere. At steady state, atmospheric $pCO_2$ is in equilibrium with the





concentration of molecular dissolved $CO_2$ (so called "aqueous $CO_2$" or $CO_2aq$) in the mixed layer of the ocean. This is

formulated by Henry's Law. $CO_2aq$ itself is a function of ocean pH, which is controlled by the ratio of DIC/total alkalinity

(TA) entering the oceans via rivers.

Alkalinity is added to the ocean via rivers that transport the weathering products of rock (cations). Today, about 0.5 to 1 Gt of

$CO_2$ is globally removed by silicate rock weathering, according to the generalised chemical formulae ((Gaillardet et al., 1999;

Moon et al., 2014):

$(Ca,Mg)SiO_3 + 2CO_2 + 2H_2O => (Ca,Mg)^{2+} + 2HCO_3^- + SiO_2 + 2H^+ => (Ca,Mg)CO_3 + SiO_2 + CO_2 + 2H_2O$

The magnitude of the alkalinity flux is determined by the silicate weathering feedback (Walker, 1981; Berner et al., 1983),

mostly by the hydrological cycle, the temperature and by the 'weatherability' of the Earth's surface. The latter is determined

by a combination of factors that make silicate rocks susceptible to chemical weathering such as uplift rate, biological alteration,

and the composition of rock exposed at the Earth's surface (Kump and Arthur, 1997; West et al., 2005).

The chemical weathering response on Earth ranges between supply/transport-limitation or weathering/kinetic-limitation,

where a limit on weathering is imposed by $pCO_2$, temperature, and hydrology, irrespective of supply. While Earth as a whole

can be considered to have a weathering-limited regime, there are many environments that approach supply limitation –

characterised by low relief and with well-developed, compositionally mature soil profiles (West et al., 2005; Maher and

Chamberlain, 2014; Isson et al., 2020), such as arable land.

In addition to input fluxes, oceans are the site of major organic and inorganic carbon export. For at least the last 500 million

years in Earth history, calcification by organisms has led to the accumulation of carbonate sediments in the ocean. Because the

dissolution of calcium carbonate is pressure- (i.e. depth-) dependent, carbonates are only preserved above the so-called

"carbonate compensation depth" (or CCD). The deep ocean below the CCD is devoid of carbonate. As the residence time of

calcium is much longer than the turnover time of the ocean (ca. 1000 yrs), it is uniformly distributed with depth. Hence, the

depth of the CCD is only determined by the carbonate ion concentration. By net dissolving or building up sediments, i.e.

shoaling or deepening of the CCD, the ocean can buffer carbon perturbations on timescales of a couple of ocean "over-

turnings" i.e. a couple of thousand years. On these timescales, atmospheric $pCO_2$ is controlled by the "Rain Ratio", i.e. the

vertical fluxes of organic carbon to calcite to the deep sea (Archer and Maier-Reimer, 1994). On timescales longer than the

biosphere and surface ocean response, rock weathering and ocean alkalization will be the primary way anthropogenic $CO_2$ will

be consumed and stored, and ocean chemistry rebalanced over the next 50+ yrs (Archer et al., 2009).

The anthropogenic carbon emissions into the atmosphere are now exceeding the natural controls as the ocean mechanism

("carbonate compensation") cannot keep pace with the anthropogenic perturbation. As a consequence, atmospheric $pCO_2$ is

steadily increasing with a seasonal modulation driven by the terrestrial biosphere, as impressively shown by the Mauna Loa

$pCO_2$ record that started in March 1958!



One of the most promising ways of taking $CO_2$ out of the atmosphere is to add alkalinity to the ocean, either directly ("Ocean Alkalinity Enhancement" or OAE) or indirectly via river run-off (produced as a consequence of EW on land), provided that alkalinity enriched waters are in contact with the atmosphere long enough for carbon drawdown to occur.

## 10 Appendix 2: The concept of (ocean) alkalinity in a nutshell

The concept of alkalinity in relation to seawater has its roots in the late nineteenth century investigations by Tornøe and Dittmar
(see: Dickson, 1992) and since then has almost exclusively been used in the context of the oceanic carbonate chemistry, for instance, with regard to the natural control of atmospheric $CO_2$. Of the 6 parameters defining the carbonate system (DIC, $CO_2aq$, $HCO_3^-$, $CO_3^{2-}$, pH & TA) only DIC (dissolved inorganic carbon) and TA (total alkalinity) are conservative, i.e. they do not change with temperature or pressure and obey a linear mixing relationship (Wolf-Gladrow et al., 2007).

In formal terms, TA is defined as the excess of proton acceptors over proton donors and the alkalinity equivalence point is
where TA=0. TA of a water sample can thus be estimated as the amount of HCl that has to be added to reach the equivalence point. The pH at which proton donors exactly balance proton acceptors (TA = 0) is defined as the so-called proton condition and denoted by $pH_0$. The proton condition is different for different acid-base systems and the choice of a particular acid-base system therefore defines the so-called "zero level of protons" for a single set of related acid-base species. The overall proton condition defines the zero level of protons for each single acid-base system.

For SW, a mixture of acid-base systems, zero level of protons and the pH where TA=0 must be defined. Hence, a single pK value must be defined/chosen which applies for all acid-base systems as a ***dividing point*** between proton acceptors and proton donors. This pK is denoted as $pK_{zlp}$, the pH at zero level protons and is specified as follows: The chemical species which dominates (largest concentration compared to the other chemical species of the same acid-base system) at $pH=pK_{zlp}$ defines the zero level of protons species, acids with $pK < pK_{zlp}$ are proton donors, and bases formed from weak acids with $pK > K_{zlp}$
are proton acceptors. Dickson (1981) chose $pK_{zlp}$ such that it is below $pK_1$ of the carbonate chemistry and so that systems that are of no interest do not contribute to TA (this was achieved by choosing $pK_{zlp}$ above those of hydrogen sulphate (pK=2) and hydrogen fluoride ($pK_{HF}$=3.2)), i.e. $pK_{HF} = 3.2 < pK_{zlp} < pK_1 = 6.3$. $pK_{zlp} = 4.5$ was chosen to correspond roughly to the pH of the conventional alkalinity end-point, and is nearly in the middle of this range between 3.2 and 6.3.

In more simple terms and as an approximation, TA can be understood as the difference between the sum of the cations minus
the anions from strong, i.e. fully dissociated, acids and bases in the ocean. If 1 kg of sea water evaporates, approximately 35 grams of salt will remain. A charge balance of the remaining cations minus anions will show a slight excess of positive charge. As charge balance is always obeyed in sea water, this excess positive charge is balanced by the dissociation of weak acids and bases (see: Wolf-Gladrow et al., 2007). To determine the alkalinity in seawater, Dickson (1981) developed an operational definition using volumetric titration with a strong acid. By far the dominant contributor to alkalinity comes from deprotonated
carbonic acid ("carbonate alkalinity") and the remainder from deprotonated boric acid[10] and a negligible small fraction from

---

[10] Note that only ca. 3.5% of TA in open ocean surface water is due to "borate alkalinity"





silicate and phosphate ("nutrient alkalinity"). The organic fraction of TA (OrgAlk, from organic acids/bases) is typically deemed negligible in the open ocean. Although it is not explicitly accounted for in conventional TA expressions, it is included by the ellipses ("...") in the equation below that stand for additional, as yet unidentified, acid-base species. Hence, a volumetric titration of seawater with a strong acid yields a TA, which consists almost completely of carbonate alkalinity. Total alkalinity

(TA) has been operationally defined by Dickson (1981):

$$TA = [HCO_3^-] + 2[CO_3^{2-}] + [B(OH)_4^-] + [OH^-] + [HPO_4^{2-}] + 2[PO_4^{3-}] + [H_3SiO_4^-] + [NH_3] + [HS^-] + \ldots - [H^+] - [HSO_4^-] - [HF] - [H_3PO_4] - [HNO_2] + \ldots$$

where square brackets denote concentrations in gravimetric units (mol kg$^{-1}$). The term "$- [HNO_2]$" was later added by Dickson (Wolf-Gladrow et al., 2007). This expression is used/applied to measure TA by titration with a strong acid (HCl); therefore, it is also called titration alkalinity.

Wolf-Gladrow et al. (2007) combined Dickson's expression for TA with the concept of electroneutrality of aquatic solutions, to derive the 'explicitly conservative' expression for TA. In a first step, the sum of concentrations of all ion, weighted by their

charges, equals zero:

$$[Na^+] + 2[Mg^{2+}] + 2[Ca^{2+}] + [K^+] + 2[Sr^{2+}] + \ldots + [H^+] + [NH_4^+] + \ldots - [Cl^-] - [Br^-] - 2[SO_4^{2-}] - [NO_3^-] - [NO_2^-] \ldots - [HCO_3^-] - 2[CO_3^{2-}] - [B(OH)_4^-] - [OH^-] - [HS^-] - [H_3SiO_4^-] - [HSO_4^-] - [F^-] - [H_2PO_4^-] - 2[HPO_4^{2-}] - 3[PO_4^{3-}] = 0$$

Next, the terms that show up in Dickson's TA expression are shifted to the right-hand side:

$$[Na^+] + 2[Mg^{2+}] + 2[Ca^{2+}] + [K^+] + 2[Sr^{2+}] + \ldots + [NH_4^+] + \ldots - [Cl^-] - [Br^-] - [NO_3^-] - [NO_2^-] - [F^-] - 2[HSO_4^-] - 2[SO_4^{2-}] - [H_2PO_4^-] - [HPO_4^{2-}] - [PO_4^{3-}] = [HCO_3^-] + 2[CO_3^{2-}] + [B(OH)_4^-] + [OH^-] + [HPO_4^{2-}] + 2[PO_4^{3-}] + [H_3SiO_4^-] + [HS^-] + \ldots [H^+] - [HSO_4^-]$$


Add electrically neutral compounds (in bold) on both sides in order to obtain Dickson's TA expression on the right-hand side:

$$[Na^+] + 2[Mg^{2+}] + 2[Ca^{2+}] + [K^+] + 2[Sr^{2+}] + \ldots + [NH_4^+] + \ldots - [Cl^-] - [Br^-] - [NO_3^-] - [NO_2^-] - \mathbf{[HNO_2]} - [F^-] - \mathbf{[HF]} - 2[HSO_4^-] - 2[SO_4^{2-}] - \mathbf{[H_3PO_4]} - [H_2PO_4^-] - [HPO_4^{2-}] - [PO_4^{3-}] + \mathbf{[NH_3]} = [HCO_3^-] + 2[CO_3^{2-}] + [B(OH)_4^-] + [OH^-] +$$

$$[HPO_4^{2-}] + 2[PO_4^{3-}] + [H_3SiO_4^-] + \mathbf{[NH_3]} + [HS^-] + \ldots - [H^+] - [HSO_4^-] - \mathbf{[HF]} - \mathbf{[H_3PO_4]} - \mathbf{[HNO_2]} + \ldots$$

The right-hand side now equals Dickson's expression of TA. In a last step, sum up the species of the same acid-base system on the left-hand side (as the total concentrations are conservative):



$[Na^+] + 2[Mg^{2+}] + 2[Ca^{2+}] + [K^+] + 2[Sr^{2+}] + \ldots - [Cl^-] - [Br^-] - [NO_3^-] - \ldots - TPO_4 + TNH_3 - 2TSO_4 - THF - THNO_2 = TA_{ec}$

where $TPO_4$ = total phosphate = $[H_3PO_4] + [H_2PO_4^-] + [HPO_4^{2-}] + [PO_4^{3-}]$, $TNH_3$ = total ammonia = $[NH_3] + [NH_4^+]$, $TSO_4$ = total sulphate = $[SO_4^{2-}] + [HSO_4^-]$, THF = total fluoride = $[F^-] - [HF]$ and, $THNO_2$ = total nitrite = $[NO_2^-] - [HNO_2]$,

respectively. Again, the ellipses stand for additional, as yet unidentified, acid-base species. This expression is called 'explicit conservative' because each single term is conservative in the sense that it does not change in a closed system with temperature or pressure and that they obey linear mixing relationships. Using $TA_{ec}$, it is easy to calculate the change of TA for dissolution or precipitation of $CaCO_3$: TA changes by 2 moles per mol of $CaCO_3$. Middelburg et al. (2020) argue that "it is important to distinguish between measurable titration alkalinity and charge balance alkalinity that is used to quantify calcification and

carbonate dissolution and needed to understand the impact of biogeochemical processes on components of the carbon dioxide system." While they refer to the ocean, the same is true for porewaters in an agricultural setting.

The contributions of other weak acid-base systems (the ellipses in Dickson's expression and in the $TA_{ec}$ expression) are minor in open ocean seawater. However, in soil pore waters, rivers, and estuaries contributions by organic acids may contribute non negligible amounts to total alkalinity. The estimation of this organic alkalinity is a challenge because the chemical composition

of the organic acids is usually not known and their pK-values vary over a wide pH range. Special titration methods (including back-titration with a strong base) have been proposed to obtain at least a rough estimate of the contribution of organic alkalinity (Wang and Cai, 2025). However, no routine analytical procedure has been developed for seawater thus far, let alone for soils.

## 11 Appendix 3: Alkalinity in soils

In comparison to the alkalinity in oceans, boron alkalinity in arable soils can be neglected as the majority of the world's

agricultural soils contain only 5–30 ppm total boron[11] (Kloke, 1980). However, "nutrient alkalinity" and "organic alkalinity[12]" can play an important role. Similar to seawater, the proton condition and $pH_0$ for soils needs to be evaluated. Defining/choosing a single pK value applying to all acid-base systems in soils as a ***dividing point*** between proton acceptors and proton donors is not trivial as dissolved humic substances are a mixture of many molecules/compounds and therefore display a range of pK values (see e.g. Fig. 5 in Fukushima et al., 1995).

The charge contribution of dissolved humic substances, i.e. humic and fulvic acids, dominantly comes from carboxylic and phenolic functional groups (Ritchie and Perdue, 2003). These have average pK values of ca. $3.7 \pm 2.4$ and ca. $12.5 \pm 1.8$, respectively (Perdue et al., 1984). As a result, for a titration to a pH of 4.5, which is the endpoint in most Gran titrations, humic acids could contribute between ca. 1.0-2.0 eq kg$^{-1}$ and fulvic acids between ca. 1.5-2.5 eq kg$^{-1}$ of negative charge, depending on initial soil pH and ionic strength of the soil solution (Milne et al., 2001). In agricultural soils, pore water DOC varies

---

[11] In addition, pKa is above 9 at a salinity of 0.
[12] Whether or not, and to what extent, organic acid-base systems contribute to titration alkalinity depends on the pK value of the organic substance!





between 12 and 104 mg C L$^{-1}$, with a median value of 33 mg C L$^{-1}$ (De Troyer et al, 2014). Assuming 50% of C in humic substances, this is equivalent to 66 mg DOM L$^{-1}$. Recent work in two agricultural soils has shown that circa 83% of pore water DOM is reactive, with the vast majority present as fulvic acids (Qin et al., 2024). This means that for a titration endpoint to pH = 4.5, the charge contribution from fulvic acids is approximately 82.5-137.5 µeq L$^{-1}$, while the charge contribution from humic acids is negligible. This back-of-the-envelope calculation shows that at an endpoint of pH 4.5, the contribution of

dissolved humic substances, specifically fulvic acids, to alkalinity is likely to be substantial. While phenolic functional groups of humic substances are included in the "titration alkalinity", (most of) the carboxylic functional groups are not. In addition, this calculation does not yet take into account the contribution of Low Molecular Weight Organic Acids (LMWOAs) that are commonly found in the rhizosphere. The bottom line is that agricultural soils are a source of organic alkalinity that, when exported and long-lived, can even be traced in coastal waters.

As argued above for seawater, charge balance is always obeyed in soils and positive charge of the cations released from EW of basalt has to be balanced by the dissociation of weak acids and bases. Hence, following the fate of the cations seems to be the method of choice to monitor alkalinity additions.

## 12 Appendix 4: Impact of agricultural practices on the CDR of EW

Here we highlight the impact of agricultural practices on the CDR of EW. Notably, liming and mineral and nitrogen
fertilization.

### 12.1 Liming

The application of lime from limestone (CaCO$_3$) adds two moles of alkalinity for one mole of DIC and therefore is an overall sink of carbon as the calcium will become charge balanced. However, if the lime is produced via calcination, which requires a lot of heat/energy, it will be an overall source of CO$_2$. Hence, the impact depends on the origin. However, most Ag-lime
originates from limestone and only negligible amounts from calcination.
Depending on soil pH and the contribution of strong acids from nitrification of ammonium fertilisers, liming can act as a carbon source or sink (West and Mcbride, 2005; Oh and Raymond, 2006; Hamilton et al., 2007).

### 12.2 Fertilisation

The main nitrogen fertiliser in agriculture is urea, but in some instances artificial fertiliser is used. If charge is added with this
fertiliser (e.g. NaNO$_3$), the NO$_3^-$ does not compete for cations released from basalt and its impact is neutral. However, most artificial N fertilisers are ammonium (NH$_4^+$) based and produced via the Haber-Bosch reaction, responsible for 1.4% of the global CO$_2$ emissions and 1% of the global energy consumption (Capdevila-Cortada, 2019). Such a fertiliser has a carbon-debt to start with and nitrification can turn NH$_4^+$ into nitric acid (HNO$_3$) where the NO$_3^-$ will compete with carbonic acid for neutralising cations. Hence, one can argue that the impact of nitrogen fertilisation in agriculture is reducing the CDR potential



of EW. In simple terms, adding acid to the soil, will drive $CO_2$ production/evasion somewhere along the chain from field to
the coastal ocean:

$$HCO_3^- + H^+ \rightarrow CO_2 + H_2O$$

However, if lime is applied in addition to fertiliser, local charge balance should be considered as part of the CDR on the
catchment scale. The overall carbon storage of the combined processes, relative to without liming, will be increased because
the introduced $Ca^{2+}$ will charge-balance nitrate export. Hence, acidity interacting with bicarbonate (from carbonate or silicate
weathering) in the field avoids carbon degassing downstream because it has been removed by the weathering process. Note,
that it will not avoid "additional" degassing due to nitric acid but we argue that it does not abate the CDR potential from EW.

A similar argument was made by Bertagni and Porporato (2022). Based on integrated hydrological and biogeochemical theory
they conclude that the alkalinization carbon-capture efficiency (ACE) across the aquatic continuum from land to ocean may
vary significantly depending on pH but do not affect the final efficiency in the ocean.

Production of $N_2O$ results from incomplete denitrification, a process which can be substantial in grasslands, where water tables
are high and anoxic conditions may prevail. $N_2O$ has a global warming potential that is nearly 300 times that of carbon dioxide

over a 100-year time scale. This should be further investigated and accounted for in carbon equivalents. It has been suggested
that EW may limit the efflux of $N_2O$ (Chiaravalloti et al., 2023) by enhancing plant N uptake. Complete denitrification
produces alkalinity and charge-balancing this through the carbonate system will result in carbon removal. The net reaction is:

$$2CaCO_3 + 4HNO_3 + 2CO_2 \rightarrow 2Ca^{2+} + 4HCO_3^- + 2N_2^{\uparrow} + 5O_2^{\uparrow}$$


Even if the timescales over which this unfolds are uncertain, carbon removal will increase despite strong acid dissolution of
feedstock. On longer timescales one can also argue that, once in the ocean, nitrogen will be incorporated in Redfield ratio into
organic matter (106 carbon for 16 nitrogen), i.e. more than 6 carbon captured for each nitrogen!

**12.3 Analogy to Terra Preta**

Amazonian black earth or "terra preta" is typically associated with human occupation, but it is uncertain whether it was created
intentionally. Interestingly, "terra preta" is not only characterised by its high organic carbon content and the presence of biochar
but also by a high density of broken ceramic artefacts (e.g. Schmidt et al., 2023). Ancient Amazonians produced a lot of
breakable pottery and the prevalence of pottery shards may be incidental and not part of a plan to improve soil fertility, but the
practice of adding biochar and pottery shards to organic leftovers of manioc, cassava, corn, papaya and bananas, sequestered

and stored carbon in the soil for centuries(Lima et al., 2002; Costa et al., 2004) and may demonstrate that broken pottery is an
essential ingredient in terra preta analogous to clay minerals for building up SOC in arable soils.



Interestingly, Anthony et al. (2025) show that the combined ground rock, compost, and biochar amendment had the greatest increases in soil C stocks over 3 years. This can be a coincidence but that composition is very similar to terra preta (except that pottery shards are replaced by crushed rock forming clay minerals). Could it be that the combination of organic amendments,

biochar and crushed rock is also the best practice for CDR?

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
