# Peer review of "Reviews and syntheses: Carbon vs. cation based MRV of Enhanced Rock Weathering and the issue of soil organic carbon."

_EGUsphere, 2025_

## Author Comment (AC2)

Figure caption: Cations released from rock feedstock (cation alkalinity) is balanced by the sum of carbonate alkalinity, organic alkalinity.and e.g. nitrate from fertilizer nitrification. Relative contributions are indicated by the thickness of the arrows. Most of the cations will be incorporated into the crop and clay minerals, which stabilize soil organic matter as a transient pool of MAOM (Mineral Associated Organic Matter).

---

## Author Response (AR2)

**point-by-point response to RC1:**

1. In Section 2: Enhanced Rock Weathering and agriculture (Lines 90-91) — I would add additional potential co-benefits of soil health such as increased water infiltration and holding capacity.

We have added the following paragraph after line 92: "Besides the potential to contribute significantly to net-zero greenhouse gas emissions goals, EW has been shown to increase water infiltration and holding capacity as additional potential agronomical benefits (e.g. Beerling et al., 2024). A recent study in a Kenyan smallholder system showed that EW boosted maize yields by over 70%, improved soil fertility, and increased micronutrient availability (Haque et al., 2025). This study demonstrates that EW is not merely a theoretical carbon removal strategy but a practical agricultural intervention with the potential to significantly improve livelihoods in some of the world's most vulnerable regions."

2. In Section 2: Enhanced Rock Weathering and agriculture (Lines 107-117) — for clarity for the 'less-well-initiated in ERW' readers, consider providing links between the role of organic matter breakdown leading to soil acidity and impact on soil porewater carbonate chemistry, $CO_2$ degassing, and the influence of basic cations on pH via charge balance as it will tie better into the readers understanding of the fate of all weathering products in the soil and how it relates to delays in CDR. Addressing this briefly in this section will also make a clearer link for some readers as to why cation- rather than carbon-accounting is preferable for estimating/verifying CDR. I appreciate this is addressed subsequently in this section and in Appendix 3, but a couple of additional sentences that makes these mechanistic linkages here rather than later (Lines 144-150) in Section 2 would be useful.

We have added the following paragraph before line 110: "High efficiency crop production is desirable, where as much biomass per acre is grown in the shortest possible time. As a consequence of root respiration, the production of humic substances, intense microbial respiration and fertilizer oxidation, soil waters become more acidic. The protons of dissociating organic or non-carbon inorganic acids may protonate bicarbonate thereby shifting the carbonate equilibrium towards a degassing of $CO_2$. The decomposition of organic matter not only releases $CO_2$, but is also an important step in the formation of MAOM as the negative charge of the remaining organic carbon will retain the basic cations via charge balance. Overall, these processes may stimulate further weathering but also delay the CDR. It is the balance between the release of basic cations, the production or organic acids and the decomposition of organic matter that control the shift in pH."

3. Section 4 — Impact of ERW on organic carbon cycling: what is missing here is a counterpoint regarding the role of MAOM on SOC stabilization. Mesocosm studies are beginning to show that the response of microbial ecosystems to the addition of silicate (and even more so with organic) amendments may have as large, if not larger impact on the fate of SOC stabilization.One paper by Sohng et al. in review at Global Change Biology addresses this – but I believe there are others that have come out recently.Or perhaps adding a statement acknowledging that recent/upcoming research papers document that impacts on MAOM creation and changes in microbial ecosystem responses (initially increased SOC mineralization as an alternative energy source due to stress of the silicate amendment; on the annual scale, a shift to increased microbial biomass (SOC)) are creating the SOC stabilization effects.  If the authors would like a preprint of the Sohng et al. paper, then we can share it, although other related studies have been published.

Just before line 209, we have added the notion that "Emerging evidence indicates that EW influences organic matter stabilization not only through added reactive minerals but also via changes in microbial activity (Sohng et al., 2025; Boito et al., 2025). Silicate amendments may initially stimulate SOM decomposition (as is common when pH increases), yet under some conditions may promote MAOM formation (Steinwidder et al., 2025). These responses are strongly context-dependent, and decreases in MAOM formation have also been observed (Sokol et al., 2024). Sohng et al. (2025) demonstrate the importance of quantifying trade-offs between the co-application of silicate and organic amendments on the fate of SOC stabilization.It is critical to explicitly integrate biotic and organic processes into EW assessments (Boito et al., 2025). Ultimately, the EW effect on SOC stocks will reflect a balance between shorter-term destabilization and longer-term stabilization of SOM through MAOM."

4. In Section 5 MRV in agricultural soils (primarily referring to Lines 235-240) — a further developed summary discussion here of how cation accounting is used in the context of the modified (again, stating 'how' it is modified) conservative total alkalinity (as shown in detail in Appendix 2), and a reference to the preferred methodology(ies) for measuring cations (in solution? in the soil, extractable fraction? all?) with citations to existing papers (summarizing in words what is presented as equations in Appendix 2) will make the main text more mechanistically linked to subsequent sections and in turn reinforce why typically used TA approaches won't work in agricultural fields — and therefore more useful to a broader range of readers.

We add after line 240: "In the open ocean, all cations released from dissolving feedstock would be exactly equivalent to the carbonate alkalinity as determined by Dickson's titration. As explained in section 3, it is due to differences in matrix (water vs. soil) that

organic matter is efficiently transferred through the food chain to below the mixed layer of the ocean but mostly remains in the top soil where it is broken down into POM and DOM creating a mismatch between titration ("carbonate") alkalinity and cation alkalinity that is mainly due to the presence of organic alkalinity. For determining the CDR in soils, we therefore propose soil-based mass-balance approach (e.g. Reershemius et al., 2023). In principle, all cations released from dissolving feedstock in agricultural soil are potential alkalinity to neutralize dissolved carbon in the ocean. Their fate is discussed in more detail in section 6."

5. Section 6: This is a well-presented and comprehensive discussion of the role of sorption and cation exchange in soils on CDR. That said, I think it would be additionally useful for the broader readership if a bit more background was provided to define soil characteristics such as base saturation, CEC and indicate the mechanistic linkages to each other and to soil pH and formation of secondary clays — especially in the context of the last paragraph in the section. Also, can the potential for the relative impact of CO2 sequestration by secondary clay-organic interaction in the ocean vs. the negative impact of cation sequestration by secondary clay formation in the soil system be addressed – even semi-quantitatively? I also question whether the impact of neo-formed clays is not the primary one in terms of secondary silicates in such systems. There is literature that addresses this. Adding a couple of soil-science based papers on this would make this discussion more robust.

We started section 6 with two paragraphs providing some background: "The cation exchange capacity (CEC) of a soil represents the total quantity of cations that can be reversibly adsorbed onto negatively charged surfaces, such as clay minerals and organic matter. These exchange sites can be occupied by both basic cations ($Ca^{2+}$, $Mg^{2+}$, $K^+$, $Na^+$) and acidic cations ($H^+$, $Al^{3+}$, $Fe^{3+}$, $NH_4^+$). While the cation incorporation into secondary clay minerals is mostly permanent due to isomorphic substitution and thus fixed, the CEC contributed by organic molecules increases with pH due to deprotonation of their functional groups (Weil and Brady, 2016; Blume et.al., 2018). Depending on pH, humic substances account for 50 – 90% of the CEC in soils. The base saturation (BS) reflects the fraction of the CEC occupied by base cations. Higher BS generally corresponds to higher soil pH, as these cations neutralize acidity. Together, soil pH, CEC, and base saturation control the mobility and availability of cations, influencing mineral weathering and the formation of secondary clays. These processes, in turn affect both, the fate of cations and carbon sequestration in enhanced weathering systems.

The majority of the cations from the dissolving feedstock end up as clays in the field and a small fraction of those *in-situ* neo-formed clays will be eroded downstream and finally end up in the ocean. However, this effect is entirely unquantified, and will vary hugely

locally. Hence, the primary process of $CO_2$ sequestration by ERW is the clay-organic carbon interaction of neo-formed clays and the primary research question for the efficacy of EW boils down to how much carbon is associated with clays (mol C per mol cation)?

Clay minerals transported via rivers to the ocean also bind organic carbon and when entering the ocean, organic rich deposits are formed in estuaries via a process called "flocculation". Unfortunately, it is difficult to assess the potential for the relative impact of $CO_2$ sequestration by secondary clay-organic interaction in the ocean vs. the impact of cation sequestration by *in-situ* clay formation in the soil system and binding SOC. The strong relationship between organic matter and mineral surface area in recent and ancient marine sediments suggests that adsorption of carbon compounds onto clay mineral surfaces played and plays a fundamental role in the burial and preservation of organic carbon (Hemingway et al., 2019; Kennedy and Wagner, 2011; Kennedy et al., 2002; 2014).

6. Robust overview of the different soil processes that can influence pathways of cations but based largely on the ERW community's literature. perhaps could expand to include more soil-based literature.

See reply to above. We agree that this has made the overall discussion more robust.

**Technical Corrections:**

7. Lines 137 to 143: perhaps this can be cut and possibly integrated into Appendix 2– this ocean-focused discussion is a bit distracting from the arable lands focus at this point.

We have taken that section out and added it after line 451 in Appendix C.

8. Line 232: consider replacing 'derive' with 'disentangle'.

Done.

9. Section 6 — Fate of the cations: Line 252: consider adding 'primarily due to pH-dependent negatively charged exchange sites provided by 'cation exchange capacity'. Again, for clarification for a broader readership.

Done.

10. Line 282: I believe 'ions' should be 'ion'. And I would add that the preferential ion sorption is also dependent on charge density (lyotropic (Hofmeister) series)

Done. Line 283 was changed to: "....., but the preferential ion sorption varies for different clay minerals, is dependent on the charge density (lyotropic (Hofmeister) series), depends on the absolute is concentration and is pH dependent (e.g. Farrah et al., 1980; Gislason et al., 1996; Kunz et al., 2004; Doi et al., 2020)."

The following citation was added:

Kunz, W., et al. (2004). "'Zur Lehre von der Wirkung der Salze' (about the science of the effect of salts): Franz Hofmeister's historical papers." Current Opinion in Colloid & Interface Science **9**(1): 19-37.

11. Lines 290 to 295: Discussion of influence of carboxylate and phenolate groups on cation mobility through the soil system — consider adding to a statement about how deprotonation of these functional groups can increase negatively charged exchangeable sites for basic cations as well as SOC bioavailability.

We included the role of organic molecules in CEC in the newly added part in section 6. In line 293 we added: "The presence of carboxylic and phenolic groups gives these substances the ability to form chelate complexes with ions such as $Mg^{2+}$, $Ca^{2+}$, $Fe^{2+}$, and $Fe^{3+}$ (Tipping, 1994), which is an important aspect of the biological role of humic acids in regulating bioavailability of metal ions to crops. The chelates release the cations lower in the soil profile, leading to podzol formation. But chelation is also important to speed up weathering, as it reduces secondary mineral precipitation at the mineral surface."

We also changed line 285 as the role of organic substances is now discussed when explaining CEC: "As mentioned above, the CEC complex is not composed purely of inorganic clays and cations will also adsorb to organic compounds such as humic substances."

12. Footnote on page 13: I'd add a reference or two for the reader as this is a review.

The footnote was placed in the main text and two references were added: One related to the impact of land use and mineral type on the stability of MAOM (Bramble et al., 2025) and one on the climate sensitivity of MAOM (Wu, Y., et al. (2025).

13. Appendix 2. Line 49. We have added Brantley et al. 2023 to the reference list. Ditto for the references listed in Line 464.

Assuming that the reviewer refers to Appendix 1(A) and line 459 (the 5 was missing in "49") and line 464, we added Brantley, S. L., et al. (2023). "How temperature-dependent

silicate weathering acts as Earth's geological thermostat." Science **379**(6630): 382-389.) was added to both references listed already.

14. Appendix 4: refs for role of SOC on cation sorption?

We referred to Rowley et al., 2021

**Citations added:**

Beerling, D. J., et al. (2024). "Enhanced weathering in the US Corn Belt delivers carbon removal with agronomic benefits." Proceedings of the National Academy of Sciences **121**(9): e2319436121.

Blume, H.-P., Brümmer, G. W., Fleige, H., Horn, R., Kandeler, E., Kögel-Knabner, I., Kretzschmar, R., Stahr, K., Wilke, B.-M. 2018. Scheffer/Schachtschabel Soil Science https://doi.org/10.1007/978-3-642-30942-7

Bramble, D. S. E., et al. (2025). "Land use and mineral type determine stability of newly formed mineral-associated organic matter." Communications Earth & Environment **6**(1): 415.

Brantley, S. L., et al. (2023). "How temperature-dependent silicate weathering acts as Earth's geological thermostat." Science **379**(6630): 382-389.

Haque, F., Möller, B., Sagina, S., Odhiambo, C., Ondolo, H., Thuo, N., Kamau, K. and S. Davies, 2025. Agronomic Performance of Enhanced Rock Weathering in a Tropical Smallholder System: A Maize Trial in Kenya. DOI: https://doi.org/10.70212/cdrxiv.2025410.v1

Hemingway, J. D., et al. (2019). "Mineral protection regulates long-term global preservation of natural organic carbon." nature **570**(7760): 228-231.Kennedy, M. J., et al. (2002). "Mineral Surface Control of Organic Carbon in Black Shale." Science **295**(5555): 657-660.

Kennedy, M. J. and T. Wagner (2011). "Clay mineral continental amplifier for marine carbon sequestration in a greenhouse ocean." Proceedings of the National Academy of Sciences **108**(24): 9776-9781.

Kennedy, M. J., et al. (2014). "Direct evidence for organic carbon preservation as clay-organic nanocomposites in a Devonian black shale; from deposition to diagenesis." Earth and Planetary Science Letters **388**: 59-70.

Kunz, W., et al. (2004). "'Zur Lehre von der Wirkung der Salze' (about the science of the effect of salts): Franz Hofmeister's historical papers." Current Opinion in Colloid & Interface Science **9**(1): 19-37.

Rowley, M. C., et al. (2021). "Evidence linking calcium to increased organo-mineral association in soils." Biogeochemistry **153**(3): 223-241.

Weil, Ray R, and Nyle C Brady. *The Nature and Properties of Soils, Global Edition*. 15th ed. Harlow, United Kingdom: Pearson Education, Limited, 2016. Print. ISBN: 9781292162232

Wu, Y., et al. (2025). "Warmer Climate Reduces the Carbon Storage, Stability and Saturation Levels in Forest Soils." Earth's Future **13**(2): e2024EF004988.

**Point to point reply to RC2:**

1. Some of the terminology may be opaque to readers unfamiliar with marine geochemistry. It would therefore be advisable to avoid using expressions like "cation currency based on the concept of the explicit conservative expression of total alkalinity" in the abstract, or to provide a concise explanatory note.

We fully agree with the reviewer that this sentence needs more explanation, especially for readers unfamiliar with marine geochemistry, and therefore removed the sentence from the abstract. The concept of the "explicit conservative expression of total alkalinity" is explained in the main text and more extensively dealt with in appendix 2.

2. While the advantages of the proposed cation-currency framework are convincingly articulated, the manuscript would benefit in my view from a bit more elements about its implementation. For instance, brief recommendations regarding sampling design, sampling depth, uncertainty propagation and potential pitfalls could enhance the paper's applicability."

We agree that it would be of interest to add more elements about the implementation and practicalities of a cation-based MRV, but it is beyond the scope of the current review to discuss methods for quantifying feedstock dissolution, sampling requirements for baseline and ongoing measurements, depth of analysis, plant uptake, downstream losses, etc. While protocols for the MRV of EW are already used for the voluntary market, e.g. isometric (https://registry.isometric.com/protocol/enhanced-weathering-agriculture), puro.earth (https://puro.earth/enhanced-rock-weathering) or Rainbow (https://docs.rainbowstandard.io/methodologies/enhanced-rock-weathering), the procedures are still extensively discussed, e.g. in the framework of Cascade Climate ( https://cascadeclimate.org/our-work#erw). Different approaches such as pore-water and/or soil-based analyses are still being discussed (e.g. Kantola *et al* 2023, Reershemius and Suhrhoff 2023, Reershemius et al., 2023, Clarkson et al., 2024, Suhrhoff et al., 2024, 2025a, Vienne et al., 2025) and the concepts are still evolving (e.g. Suhrhoff et al., 2025a, b).

3. Additionally, including a schematic figure contrasting the proposed approach with conventional pore-water and carbon-based approaches would substantially improve accessibility and comprehension for a broader audience.

We gladly followed the reviewer's suggestion and included a schematic figure at the end of section 2 "Enhanced Rock Weathering and agriculture":

[Figure]

Figure 1: Cations released from rock feedstock (cation alkalinity) is balanced by the sum of carbonate alkalinity, organic alkalinity.and e.g. nitrate from fertilizer nitrification. The relative contribution of the acids is indicated by the thickness of the arrows. Note that their relative importance can vary for different soils and under different conditions. Most of the cations will be incorporated into the crop and clay minerals. The latter stabilize soil organic matter as a transient pool of MAOM (Mineral Associated Organic Matter). *Created in BioRender. Reershemius, T. (2025) https://BioRender.com/9vz62gz*

4.  One open research question not explicitly discussed concerns the potential ecological feedbacks arising from the addition of large quantities of rock-flour. Such amendments may directly or indirectly affect soil fauna, which are known to play an important role in regulating SOM decomposition and stabilization. Changes in the activity or abundance of key soil ecosystem engineers, such as earthworms and other macro- and meso-decomposer groups, could have long-term and unpredictable consequences on SOM stabilization. This ecological dimension remains largely overlooked in the current ERW literature and, in my view, deserves a bit more attention.

We have added a paragraph on potential ecological feedbacks arising from the addition of large quantities of rock-flour after line 182 in section 4: "Rock amendments may directly or indirectly affect soil fauna, which are known to play an important role in regulating SOM decomposition and stabilization. Changes in the activity or abundance of key soil ecosystem engineers, such as earthworms, but also other macro- and meso-decomposer groups, could have long-term and unpredictable consequences on SOM stabilization (Lubbers et al, 2013; Zhang et al, 2013).

Earthworms are often regarded as key ecosystem engineers due to their capacity to ingest, fragment, mix and transport inorganic and organic matter, making their feeding and burrowing behaviour a key ecological property (Vidal et al., 2023). The optimal pH for earthworms ranges between 6 and 8 (Hou et al., 2005). Yet, no changes in the survival of deep-burrowing (e.g. *Lumbricus terrestris*) or shallow-burrowing (e.g. *Aporrectodea caliginosa*) earthworms, both naturally occurring in the basaltic soils on Iceland (Sigurdsson and Gudleifsson, 2013), were observed after basalt amendment, even though the pH in basalt amended mesocosms rose up to circa 8.5 (Vienne et al., 2024). In artificial organo-mineral systems, the survival and activity of the shallow-burrowing species *Apporectodea calignosa* and *Allolobophora chlorotica,* depended mostly on variables influencing the structure and drainage potential of these systems (Calogiuri et al. (2025a). These authors also found that surviving earthworms had a neutral or negative effect on weathering indicators in their artificial organo-mineral mixtures, which they attributed to the short length of their experiments or changes in organic rather than inorganic carbon concentration. In contrast, dead earthworms enhanced almost all weathering indicators considered, which was suggested to be the result of microbial processes stimulated via decomposing earthworm bodies. These results were recently confirmed by Calogiuri et al. (2025b), who showed that while live earthworms directly stimulated the formation of mineral-associated organic matter (MAOM), dead earthworms stimulated microbial activity, which enhanced both MAOM formation and inorganic carbon capture.

Interestingly, Needham et al. (2005) showed that the ingestion of unweathered basalt from Iceland by the common marine lugworm *Arenicola marina*, had no detrimental

impact on the worms but that weathering rate was one hundred to one thousand times faster than in experiments of standard, inorganic basaltic weathering. They concluded that sediment ingestion and the entire coprophagic cycle are highly significant for sediment alteration, early diagenesis and the origin of clay minerals in sedimentary rocks."

**New References added:**

Calogiuri, T., et al. (2025). "How earthworms thrive and drive silicate rock weathering in an artificial organo-mineral system." Applied Geochemistry **180**: 106271.

Calogiuri, T. et al (2025). "Alive and dead earthworms capture carbon during mineral weathering through different pathways" Communications Earth & Environment

J. Hou, Y.Qian, G. Liu and R. Dong. "The Influence of Temperature, pH and C/N Ratio on the Growth and Survival of Earthworms in Municipal Solid Waste" Agricultural Engineering International: the CIGR Ejournal. Manuscript FP 04 014. Vol. VII. November, 2005

Kantola, I. B., et al. (2023). "Improved net carbon budgets in the US Midwest through direct measured impacts of enhanced weathering." Global Change Biology **29**(24): 7012-

Lubbers, I. M., et al. (2013). "Greenhouse-gas emissions from soils increased by earthworms." Nature Climate Change **3**(3): 187-194.

Needham, S., et al. (2006). "Sediment ingestion by worms and the production of bio-clays: A study of macrobiologically enhanced weathering and early diagenetic processes." Sedimentology **53**: 567-579.

Niron, H., et al. (2025). "Alkalinity production and carbon capture from dunite weathering: Individual effects of oxalate, citrate, and EDTA salts." Chemical Engineering

Reershemius, T. and T. J. Suhrhoff (2023). "On error, uncertainty, and assumptions in calculating carbon dioxide removal rates by enhanced rock weathering in Kantola et al., 2023." Glob Chang Biol **30**(1): e17025.

Sigurdsson, B. and B. Gudleifsson (2013). "Impact of afforestation on earthworm populations in Iceland." Icelandic Agricultural Sciences **26**: 21-36.

Suhrhoff, T. J., Khan, A., Zhang, S., Woollen, B. J., Reershemius, T., Bradford, M. A., Polussa, A., Milliken, E., Raymond, P. A., and Reinhard, C. 2025a. Aggregated monitoring of enhanced weathering on agricultural lands, CDRXIV, 2025.

Suhrhoff, T. J., Reershemius, T., Jordan, J. S., Li, S., Zhang, S., Milliken, E., Kalderon-Asael, B., Ebert, Y., Nyateka, R., and Reinhard, C. 2025b. Updated framework and signal-to-noise analysis of soil mass balance approaches for quantifying enhanced weathering on managed lands, CDRXIV, 2025.

Vidal, A., et al. (2023). Chapter One - The role of earthworms in agronomy: Consensus, novel insights and remaining challenges. Advances in Agronomy. D. L. Sparks, Academic Press. **181:** 1-78.

Zhang, W., et al. (2013). "Earthworms facilitate carbon sequestration through unequal amplification of carbon stabilization compared with mineralization." Nature Communications **4**(1): 2576.